# Vibrational Effects on the Acoustic Performance of Multi-Layered Micro-Perforated Metamaterials

Cédric Maury [1] and Teresa Bravo [2,*]

[1] Laboratoire de Mécanique et d'Acoustique (UMR), Centrale Marseille, Aix Marseille University, Centre National de la Recherche Scientifique, 38 rue Frédéric Joliot-Curie, 13013 Marseille, France; cedric.maury@centrale-marseille.fr

[2] Instituto de Tecnologías Físicas y de la Información (ITEFI), Consejo Superior de Investigaciones Científicas (CSIC), Serrano 144, 28006 Madrid, Spain

* Correspondence: teresa.bravo@csic.es

**Abstract:** Broadband noise reduction over the low–mid frequency range in the building and transportation sectors requires compact lightweight sound absorbers of a typical subwavelength size. The use of multi-layered, closely spaced (micro-)perforated membranes or panels, if suitably optimized, contributes to these objectives. However, their elasticity or modal behaviors often impede the final acoustical performance of the partition. The objective of this study is to obtain insights into the vibrational effects induced by elastic limp membranes or panel volumetric modes on the optimized sound absorption properties of acoustic fishnets and functionally graded partitions (FGP). The cost-efficient global optimization of the partitions' frequency-averaged dissipation is achieved using the simulated annealing optimization method, while vibrational effects are included through an impedance translation method. A critical coupling analysis reveals how the membranes or panel vibrations redistribute the locations of the Hole-Cavity resonances, as well as their cross-coupling with the panels' first volumetric mode. It is found that elastic limp micro-perforated membranes broaden the pass-band of acoustic fishnets, while smoothing out the dissipation ripples over the FGP optimization bandwidth. Moreover, the resonance frequency of the first panels mode sets an upper limit to the broadband optimization of FGPs, up to which a high dissipation, high absorption, and low transmission can be achieved.

**Keywords:** sound absorption; micro-perforates; panels vibration; global optimization

## 1. Introduction

The design of compact and lightweight acoustic partitions with a high performance in the low-frequency range constitutes a challenging requisite within the field of noise control and vibration [1,2]. For instance, there is interest from the aeronautical industry for multi-layered fuselage partitions that could block the transmission of external wall pressures into aircraft cabins. Their low-frequency components are essentially induced by the ultra-high bypass ratio engines of modern fuel-efficient turbofans [3]. Another concern is to increase the absorption of sidewall partitions to improve the acoustical comfort inside aircraft cabins. Low-frequency noise issues also appear in the automotive industry when turbulent boundary layer noise, which peaks around 600 Hz, is transmitted into car cabins and excites its acoustic modes [4]. Strong constraints are set to use compact and lightweight materials for reducing the embarked mass and fuel consumption in surface and air transport, thus making difficult the use of classical passive or active noise control techniques.

Acoustic metamaterials have attracted attention recently as alternatives for obtaining perfect absorption or zero transmission in the low-frequency range. Interestingly, they can be designed at subwavelength scales to act either on sound absorption or transmission on selective spectral bands. For instance, the use of multi-layered identical perforated panels has been considered for impeding the sound transmission under different angles

of the incident plane wave [5–8]. The simplest unit constitutes two thick perforated panels separated by a thin air gap, denoted as a double acoustic fishnet [5]. It presents a transmission blockage at a frequency proportional to the inverse of the center-to-center distance between two holes. This has been observed for different thicknesses of panels and air gaps. However, this blockage occurs at very high frequencies, typically 17 kHz [6], due to the millimetric spacing between the holes. Moreover, it implies large reflections on the incident side, therefore providing an insulating control device that is also highly reflecting. To increase the efficiency bandwidth, a partition composed of an increasing number of equally spaced perforated plates was studied under a normal incident plane wave [7]. A set of stop-bands and pass-bands for sound transmission was analyzed in terms of the Bloch–Floquet theory, but significant sound transmission was still observed in the first pass-band at low frequencies. To improve the insulating properties, the perforated multi-layer partition was covered by an ultrafine membrane that became very rigid at low frequencies, therefore efficiently blocking the sound transmitted over a wide bandwidth, but still at the expense of large reflections [8].

To design efficient acoustic devices with simultaneous reductions in both reflection and transmission, functionally graded multilayer partitions were designed based on the Critical Coupling Condition [9,10]. This criterion aims at adjusting visco-thermal losses to exactly balance the amount of reflection and transmission leaking out of a system, so that all the incident power is fully dissipated. It enabled the design of trapping absorbers constituting of a series of chirped Helmholtz resonators [11], with gradual axial variation in the depths of the cavities. Dissipation ripples due to the individual resonances could be smoothed out by adding either porous foam on the resonators apertures [12] or by suitably tuned micro-perforated panels [13] as a more robust solution.

In this work, we aim at enhancing the bandwidth of the absorption and transmission of functionally graded materials constituting several layers of (micro-)perforated panels (MPP). The idea is to maximize the dissipation by adjusting the porosity of the panels from the incident side to the transmitting side. The novelty is to assess the vibrational effects of either elastic limp membranes or resonating thin panels on the acoustical performance of the optimized partition. Up to now, vibrational effects have essentially been addressed on single-layer micro-perforated absorbers of an infinite extent [14,15]. Several works have also analyzed the interaction between the first MPP modes and the main Helmholtz resonance [16–23], but not with a set of Helmholtz-type resonances induced by acoustic fishnets [5–7] or functionally graded multi-layered metamaterials [24–29], where the MPPs are assumed to be rigid.

In particular, structural acoustic coupling due to the flexural vibration of one-dimensional holey porous plates has been theoretically examined as a function of the perforation ratio [14], assuming panels of an infinite lateral extent. The formulation has been extended to the three-dimensional case to assess the reduction in radiated power when backing a perforated board with a honeycomb-filled cavity [15]. This model was further applied to a finite flexible MPP simply supported in a circular duct to study the panel-type absorption peaks [16]. The influence of the structural and acoustic resonances was studied for widening the absorption bandwidths of flexible MPP absorbers [17,18], eventually curved [19] with elastically restrained edges using a modal superposition method [20]. Bolton [21] analyzed how the modal behavior of a membrane is modified by the perforations. Tournadre et al. [22] used a Finite Element Method (FEM) to assess the effects of boundary conditions on a coupled vibroacoustic MPP system. FEM has also been used to find out how this absorption is modified by attaching local resonators on rigidly backed elastic MPPs [23], how the interactions cancel out between a parallel arrangement of rigidly backed elastic MPPs [30], and how micro-perforations themselves can dampen panel vibrations [31].

Section 2 presents the cost-efficient impedance translation method used to predict the acoustic properties of multi-layered MPP partitions, which can account for elastic or resonating panels. It is shown how unit dissipation peaks can be analyzed using a scattering matrix approach. Section 3 shows through parametric studies the performances of acoustic

fishnets made up of identical (micro-)perforated membranes or panels and the influence of their vibrations. Section 4 extends these parametric studies to the case of functionally graded elastic (micro-)perforates whose broadband dissipative properties are optimized by the simulated annealing method. Their high dissipation performance is analyzed using the scattering matrix approach. Finally, Section 5 discusses the vibrational effects of elastic limp membranes and thin resonant panels, especially how they favor or impede the broadband acoustical performance of acoustic fishnets or functionally graded partitions (FGPs).

## 2. Methods

The following approaches are now described: the impedance translation method (ITM) and the scattering matrix formulation. The ITM will be used for modeling the acoustical properties of multi-layered (micro-) perforated partitions. The ITM has been developed in the frame of multi-layer absorbers [32]. It overcomes the limitation of the electro-acoustic analogy that assumes each air cavity is loaded by a hard wall. It has been successfully assessed against standing wave tube measurements for triple-layered micro-perforated panels [33], as long as the thickness of the panels is much smaller than the acoustic wavelength.

The scattering matrix formulation and its eigen-analysis come from laser physics and have been applied to acoustics [9,10] to analyze two-port (reflecting and transmitting) subwavelength metamaterials made up of Helmholtz resonators. In this work, they will serve as a tool for obtaining insights into the acoustical performance of the partition. In particular, they will determine which resonant states contribute to the full dissipation of the incident disturbance by the partition.

The studied partitions consist of an arbitrary sequence of $(N+1)$ thin rigid MPPs of thickness $d_{p,n}$ separated by $N$ air gaps of depth $d_{g,n}$, as depicted in Figure 1 for an unbacked double-layer partition.

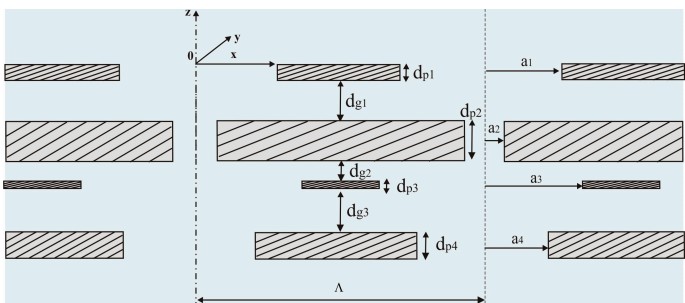

**Figure 1.** Sketch of a double-layer micro-perforated partition.

The panels are coined as micro-perforated when the radius $a_n$ of the circular perforations is lower than 0.5 mm. The pitch of the holes associated with the $n$th panel (or the center-to-center distance between the holes) is denoted as $\Lambda_n$, with $\Lambda_n > 2a_n$. For non-staggered holes, the corresponding perforation ratio reads:

$$\sigma_n = \frac{\pi a_n^2}{\Lambda_n^2} \tag{1}$$

The $N$-layered partition is assumed to be lateral to an infinite extent and undergoes normal plane wave incidence.

### 2.1. Impedance Translation Method

The ITM considers each MPP as an element characterized by its effective transfer impedance $Z_{\mathrm{MPP},n}$. Assuming a sufficient number of holes per acoustic wavelength $\lambda$,

typically $\Lambda_n < \lambda/4$, the effective transfer impedance is obtained by dividing the transfer impedance, $Z_{H,n}$, of every single hole by the perforation ratio $\sigma_n$ [34]. It reads:

$$Z_{\text{MPP},n} = \frac{Z_{H,n}}{\sigma_n}. \tag{2}$$

Maa [35] derived the transfer impedance of a single hole, assimilated to a short cylinder, by solving, in velocity, the momentum's conservation equation for a rigid tube filled with a viscous fluid and driven by a difference of pressure, $\Delta p_n$, between both sides of the tube. By imposing a zero-particle velocity at the hole walls and averaging the axial velocity over a cross-sectional area of the hole, one obtains:

$$Z_{H,n} = \frac{\Delta p_n}{\overline{v}_n} = \frac{\eta \, \text{Sh}_n}{\sqrt{2} a_n} - j\omega\rho_0\delta_n a_n - j\omega\rho_0 d_{p,n}\left[1 - F\left(\text{Sh}_n\sqrt{j}\right)\right]^{-1}, \tag{3}$$

where $\overline{v}_n$ is the averaged axial velocity and $F(x) = 2J_1(x)/[xJ_0(x)]$, $J_1$, and $J_0$ are Bessel functions of the first and zeroth orders. Note that an omitted time-dependence $e^{-j\omega t}$, with $\omega$ as the angular frequency, was assumed to derive Equation (3). $Z_{H,n}$ is expressed in terms of the Shear number $\text{Sh}_n$, the ratio between the hole radius and the in-hole viscous boundary layer thickness $\delta_v$. It reads:

$$\text{Sh}_n = \frac{a_n}{\delta_v} = a_n\sqrt{\frac{\omega\rho_0}{\eta}}, \tag{4}$$

where $\eta$ is the air dynamic viscosity and $\rho_0$ is the air density. Equation (3) can be recast as $Z_{H,n} = Z_{H-\text{ext},n} + Z_{H-\text{in},n}$, with:

$$Z_{H-\text{in},n} = -j\omega\rho_0 d_{p,n}\left[1 - F\left(\text{Sh}_n\sqrt{j}\right)\right]^{-1}, \tag{5}$$

as the inner term that accounts for the viscous dissipation and inertial effects inside the holes, and:

$$Z_{H-\text{ext},n} = \frac{\eta \, \text{Sh}_n}{\sqrt{2} a_n} - j\omega\rho_0\delta_n a_n, \tag{6}$$

which gathers the outer correction terms. The first term of Equation (6) accounts for the frictional dissipation induced by the sheared acoustic velocity at the inlet/outlet of the holes, whereas the second term describes the inertial effect induced by the mass of air pushed back and forth at both ends of the hole. It is associated with an added-length correction factor, $\delta_n = 16/(3\pi)$ [35]. If $\Lambda_n \leq 10a_n$, e.g., when $\sigma_n \geq 3$ %, the acoustic flow interaction between the holes lowers the external added-length, $\delta_n a_n$. This correction is accounted for through Fok's function $\psi_{F,n}$ [36], a monotonically decreasing function of $\sigma_n$, such that $\delta_n = 16\,\psi_F(\sigma_n)/(3\pi)$ in the general case.

In the ITM, the incident and reflected plane wave solutions are written within each air layer in terms of the acoustic pressure and particle velocity. Boundary conditions such as the jump of pressure and the continuity of the surface-averaged velocity $\overline{v}_n$ are applied across each MPP [32]. This leads to recursive expressions for the input impedance of the first panel in terms of the effective transfer impedance, $Z_{\text{MPP},1}$, of the first panel, plus the input impedance, $Z_{\text{in},1}$, of the first air gap, a function of the second-panel input impedance, and so on, until the last panel backed by the air characteristic impedance, $Z_0 = \rho_0 c_0$, with the sound speed $c_0$. The input impedances, $Z_{\text{in},n} = p_n/\overline{v}_n$, at the surfaces of each panel-cavity section ($n = 1, \ldots, N$) are thus related by [33]:

$$Z_{\text{in},n} = Z_{\text{MPP},n} + Z_0 \frac{Z_{\text{in},n+1}\cos(kd_{g,n}) + jZ_0\sin(kd_{g,n})}{Z_0\cos(kd_{g,n}) + jZ_{\text{in},n+1}\sin(kd_{g,n})}, \tag{7}$$

where $k = \omega/c_0$ is the acoustic wavenumber and $Z_{\text{in},N+1} = Z_{\text{MPP},N+1} + Z_0$ for the last transmitting panel. The transfer impedance, $Z_{n1} = p_{N+1}/\overline{v}_1$, across the multi-layer partition is obtained as [18]:

$$Z_{n1} = Z_0 \prod_{n=1}^{N} \frac{Z_0}{Z_0 \cos(kd_{g,n}) + jZ_{\text{in},n+1} \sin(kd_{g,n})}. \tag{8}$$

The absorption coefficient is defined as the fraction of the incident energy not reflected by the partition. It reads $\alpha = 1 - |r_1|^2$, with $r_1 = (Z_{\text{in},1} - Z_0)/(Z_{\text{in},1} + Z_0)$, the normal incidence reflection coefficient, and $Z_{\text{in},1}$ being obtained from Equation (7). The transmission loss (TL) is defined by $\text{TL} = -10 \log_{10}(\tau)$, in terms of $\tau = |t_{n1}|^2$, part of the incident power transmitted by the partition, with $t_{n1} = 2Z_{n1}/(Z_{\text{in},1} + Z_0)$, the normal incidence transmission coefficient. Part of the incident power not reflected nor transmitted is necessarily dissipated by visco-thermal effects through the MPP holes and within the air gaps (to a lesser extent if $d_{g,n} > \delta_v$). From the conservation of energy, the dissipated power reads $\alpha - \tau$.

### 2.2. Scattering Matrix Formulation

A useful approach to obtaining insights and finely tuning the acoustic performance of the multi-layered partition is to examine the eigenvalues/eigenvectors associated with the scattering matrix of the two-port system [9,10]. Following the notations of Figure 2, A and D denote the amplitudes of the ingoing (incident and back-transmitted) waves entering the partition, whereas B and C correspond to the amplitudes of the outgoing (reflected and transmitted) waves.

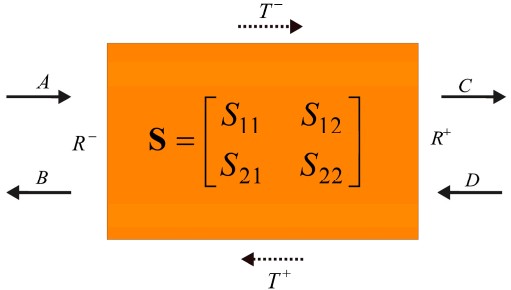

**Figure 2.** Schematic of the ingoing and outgoing waves involved in the scattering matrix of a two-port system.

The outgoing waves vector [C B] is related to the ingoing waves vector [A D] by the so-called scattering matrix **S** as follows [37]:

$$\begin{bmatrix} C \\ B \end{bmatrix} = \mathbf{S} \begin{bmatrix} A \\ D \end{bmatrix} = \begin{bmatrix} T^- A + R^+ D \\ R^- A + T^+ D \end{bmatrix}, \tag{9}$$

where $\mathbf{S} = \begin{bmatrix} T^- & R^+ \\ R^- & T^+ \end{bmatrix}$ depends on the reflection coefficients for the left ($R^-$) and right ($R^+$) incidences and the transmission coefficient $T^- = T^+ = T$. Note the invariance of the left-to-right ($T^-$) and right-to-left ($T^+$) transmission coefficients, even if the system is not symmetric, whereas $R^- = R^+$ is only valid if the system is symmetric.

The two eigenvalues of **S** read:

$$\lambda_{1,2} = T \mp \sqrt{R^+ R^-}, \tag{10}$$

and the associated eigenvectors are given by:

$$\mathbf{v}_1 = \left( R^-, -\sqrt{R^+ R^-} \right), \tag{11}$$

$$\mathbf{v}_2 = \left( R^+, \sqrt{R^+ R^-} \right), \tag{12}$$

Given a dissipative system such as an MPP partition, CCC is achieved at a specific frequency when all the incident energy entering the system is dissipated by visco-thermal effects. It results in the absence of outgoing waves, so that Equation (9) reads:

$$\mathbf{S} \begin{bmatrix} A \\ D \end{bmatrix} = \begin{bmatrix} 0 \\ 0 \end{bmatrix}. \tag{13}$$

Equivalently, CCC can be formulated as the following eigenvalue problem:

$$\mathbf{S}\mathbf{v}_{1,2} = \lambda_{1,2}\mathbf{v}_{1,2} = \mathbf{0}, \tag{14}$$

where $\lambda_{1,2} = 0$, e.g., when $T = \pm\sqrt{R^+ R^-}$. Hence, CCC is achieved when $\lambda_{1,2}(f) = 0$, e.g., when the two eigenvalues are zero-valued at the same real frequency. Previous works [10] have further considered a complex frequency variable, $f = f_r + \mathrm{j} f_i$, to account for losses in the system. CCC is then achieved in the complex frequency plane when $|\lambda_{1,2}(f_r + \mathrm{j} f_i)| = 0$ on the real frequency axis ($f_i = 0$). This approach will be useful in Section 4.3 for examining the distribution of losses amongst the resonances of the optimized FGPs.

## 3. Acoustic Performance of Multi-Layered Identical MPPs

The ITM is used to predict the acoustical performance (dissipation, absorption, and transmission under normal incidence) of an acoustic fishnet with an overall depth 43 mm, made up of six identical rigid MPPs with a thickness 0.5 mm and a hole pitch of 8 mm, separated by small air gaps of 8 mm ($d_{g,n} = \Lambda_n = \Lambda$). Its performance is examined in Figures 3–5 over a broad range of frequencies from 10 Hz to 7 kHz, e.g., from $\Lambda/\lambda = 2.3 \times 10^{-4}$ to 0.1633, when varying the radius of each MPP hole from 0.1 mm to 3.9 mm, e.g., from $a/\Lambda = 0.0125$ to 0.4875 in units of $\Lambda$. Note that the model of Maa given by Equations (2)–(6) for the effective transfer impedance of the MPPs fails to be valid if there are less than four holes per wavelength, e.g., when $\Lambda/\lambda > 0.25$.

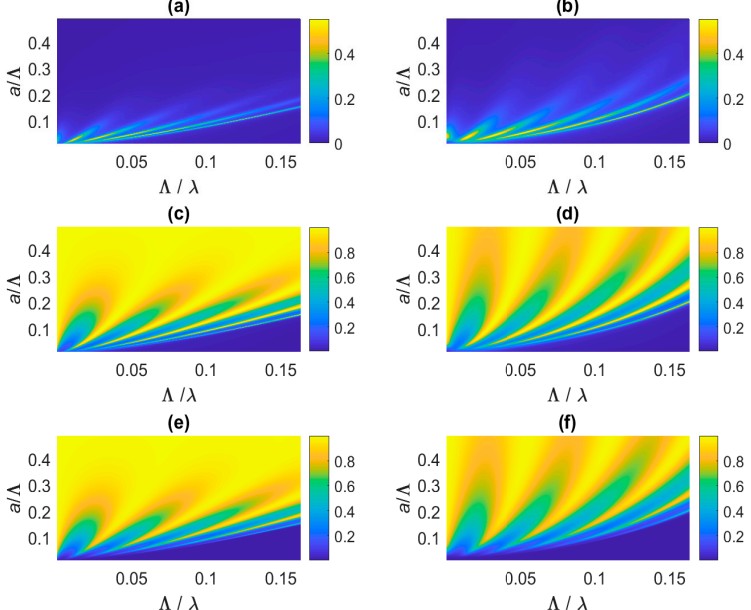

**Figure 3.** Dissipation (**a**,**b**); absorption (**c**,**d**); and transmission (**e**,**f**) coefficients of a 5-layer acoustic fishnet as a function of the radius of the holes and of the spatial frequency (inverse of the acoustic wavelength), non-dimensionalized with respect to the pitch of the holes: simulations by the enhanced modal matching method (**left column**) and by the impedance translation method (**right column**).

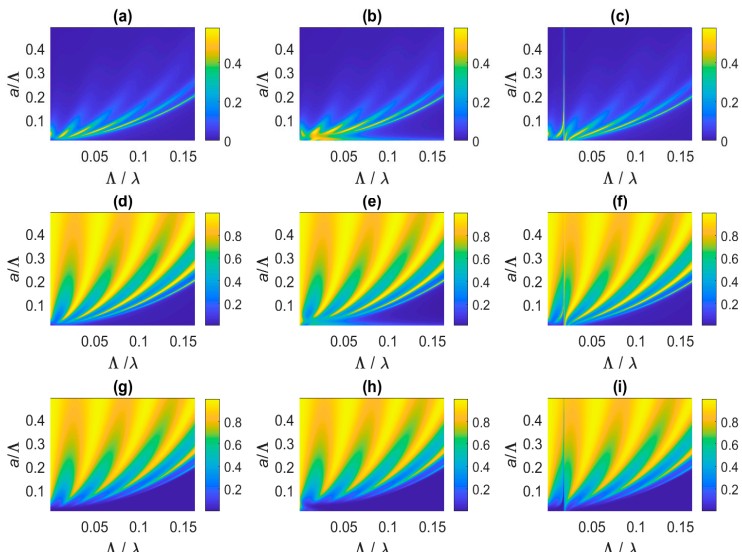

**Figure 4.** Dissipation (**a**–**c**); absorption (**d**–**f**); and transmission (**g**–**i**) coefficients of a 5-layer acoustic fishnet as a function of the radius of the holes and of the spatial frequency (inverse of the acoustic wavelength), non-dimensionalized with respect to the pitch of the holes, assuming rigid panels (**left column**), elastic limp membranes (**mid column**), and resonant vibrating panels (**right column**) when simulated by the impedance translation method.

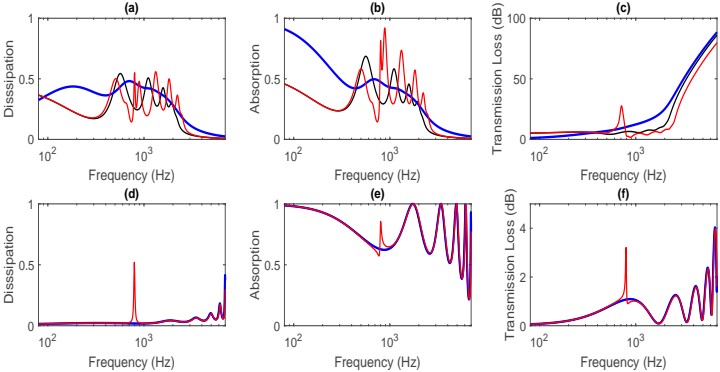

**Figure 5.** Dissipation (**a**,**d**); absorption (**b**,**e**); and transmission loss (**c**,**f**) spectra of a 5-layer acoustic fishnet made up of micro-perforates (top, $a/\Lambda = 0.04$, $a = 0.3$ mm) or perforates (bottom, $a/\Lambda = 0.2$, $a = 1.6$ mm) as a function of frequency, assuming rigid panels (black), elastic limp membranes (blue), and resonant vibrating panels (red), all simulated by the impedance translation method.

### 3.1. The Acoustic Fishnet Performance

One observes in Figure 3 a pass-band pattern (in yellow) whose bandwidth broadens when $a/\Lambda$ increases. For instance, the pass-band extends up to $\Lambda/\lambda = 0.08$ (3400 Hz) as long as $a/\Lambda$ increases up to 0.07 ($a = 0.56$ mm). In addition, $\Lambda/\lambda$ ranges up to 0.15 (6375 Hz) as long as $a/\Lambda$ increases up to 0.16 ($a = 1.3$ mm). Figure 3b,d,f show that, if $\Lambda/\lambda$ is increased beyond these limits, the acoustic fishnet exhibits a stop-band (in blue) with total reflection, no transmission, and no dissipation. This band gap pattern is typical of periodic multi-layered MPPs [5–7]. The sequence of six rigid MPPs typically leads to five Hole-Cavity (HC) resonances in the pass-band. These HC resonances appear through narrow peaks of dissipation and wide peaks of absorption and transmission. These resonances are due to the in-phase air mass in the MPP holes that couples with the air stiffness in the gaps. Within the pass-band, their resonance frequencies increase when the ratio $a/\Lambda$ increases, e.g., when the perforation $\sigma_n$ increases (see Equation (1)). Indeed, from Equations (2) and (3), the effective mass in the holes, $\mathrm{Imag}(Z_{\mathrm{MPP},n})$, decreases as $1/\sigma_n$, so that the HC resonance frequencies increase when $a/\Lambda$ increases.

Comparing Figure 3a,c,e obtained from the enhanced modal matching (EMM) method [38], respectively, to Figure 3b,d,f obtained from the ITM shows a very good correlation, as long as $a/\Lambda < 0.13$ ($a < 0.5$ mm), e.g., for a partition made up of micro-perforated panels. For perforated partitions ($a > 0.5$ mm), the trends are still captured by the ITM, although with a dependence of the HC resonance frequencies with respect to $a/\Lambda$, all become less linear as the panels' porosity increases. Indeed, the EMM, validated against FEM visco-thermal modeling of multi-layered MPP partitions [38], fully accounts for the non-planar in-hole modes and evanescent waves in the air gaps, whereas only normal plane waves are assumed in the ITM. The latter method is computationally much less expensive than the former, and thus more conducive to optimization studies.

### 3.2. Influence of the MPP Vibrations on the Acoustic Fishnet Performance

To describe the effect of the MPP vibrations on the acoustical performance of the multi-layered partition, the ITM is extended to account for thin elastic limp micro-perforated membranes and the first volumetric mode of the MPPs. Under normal incidence, the transfer impedance of the $n$th elastic (non-perforated) membrane of density $\rho_{p,n}$ and normal velocity $v_{m,n}$ is reduced to:

$$Z_{m,n} = \frac{\Delta p_n}{v_{m,n}} = -\mathrm{j}\omega\,\rho_{p,n}d_{p,n}, \tag{15}$$

Considering a micro-perforated membrane that vibrates due to the pressure difference $\Delta p_n$ between the holes' apertures, the acoustic particle velocity at the wall of each hole is not zero, unlike what was assumed in the case of rigid MPPs, but it equates to a membrane normal velocity $v_{m,n}$. As in Section 2.1, one solves, in velocity, the momentum's conservation equation assuming a vibrating tube filled with a viscous fluid and driven by a difference of pressure $\Delta p_n$. One then imposes a non-zero boundary condition for the velocity at the walls of each hole. Averaging the axial velocity over the cross-sectional area of each hole and dividing it by $\sigma_n$ provides the following expression for the effective transfer impedance of a thin elastic micro-perforated membrane [39]:

$$Z'_{\mathrm{MPP},n} = Z_{\mathrm{MPP-ext},n} + \left\{\frac{F(\mathrm{Sh}_n\sqrt{\mathrm{j}})}{Z_{m,n}} + \frac{1}{Z_{\mathrm{MPP-in},n}}\right\}^{-1}, \tag{16}$$

where $Z_{\mathrm{MPP-ext},n} = Z_{\mathrm{H-ext},n}/\sigma_n$ and $Z_{\mathrm{MPP-in},n} = Z_{\mathrm{H-in},n}/\sigma_n$. $Z_{\mathrm{H-in},n}$, $Z_{\mathrm{H-ext},n}$ and $Z_{m,n}$ are, respectively, given by Equations (5), (6) and (15).

To account for the effect of the first volumetric resonance of the vibrating panel in the effective transfer impedance of the MPPs, one assumes parallel coupling [40] between the impedance of the rigid MPP, $Z_{\mathrm{MPP},n}$, and that of the first mode resonance $\widetilde{Z}_{p,n}$. It results in the modified overall transfer impedance:

$$\widetilde{Z}_{\mathrm{MPP},n} = \frac{\Delta p_n}{v_{\mathrm{rel},n}} = \frac{Z_{\mathrm{MPP},n}\widetilde{Z}_{p,n}}{Z_{\mathrm{MPP},n} + \widetilde{Z}_{p,n}}, \tag{17}$$

where the first modal impedance of the $n$th panel is given by [41];

$$\widetilde{Z}_{p,n} = \frac{\Delta p_n}{v_{\mathrm{mod},n}} = \frac{\mathrm{j}\rho_{p,n}d_{p,n}N_{1,p,n}}{\omega}\left(\omega_{1,p,n}^2 - \omega^2 - 2\mathrm{j}\xi_{1,p,n}\omega_{1,p,n}\omega\right), \tag{18}$$

where $N_{1,p,n}$ is the squared norm of the mode, $\xi_{1,p,n}$ is the damping ratio of the first-panel mode, and $\omega_{1,p,n} = 2\pi f_{1,p,n}$ is the angular resonance frequency of the mode. Equation (17) is obtained after substituting the effective averaged velocity inside the MPP holes, $\sigma_n\bar{v}_n = \Delta p_n/Z_{\mathrm{MPP},n}$, and the panel modal velocity, $v_{\mathrm{mod},n} = \Delta p_n/\widetilde{Z}_{p,n}$, into the relative velocity between the air particle inside the holes and the vibrating panel, $v_{\mathrm{rel},n} = \sigma_n\bar{v}_n - v_{\mathrm{mod},n}$.

Figure 4 shows the effect of elastic membranes and resonant vibrating panels on the acoustic performance (dissipation, absorption, and transmission) of the five-layer acoustic fishnet for a broad range of variations in $(a, \lambda)$ in units of $\Lambda$. Figure 5 shows these effects assuming either micro-perforated panels with $a/\Lambda = 0.04$ or perforated panels with $a/\Lambda = 0.2$. The elastic limp membranes have a surface density of 0.08 kg/m$^2$, typical of low-density polyethylene films, whereas the resonant vibrating panels in aluminum have a surface density of 1.25 kg/m$^2$, so the resonance frequency of the first volumetric mode is $f_{1,p,n} = 800$ Hz. A damping ratio of $\xi_{1,p,n} = 1\%$ is chosen and $N_{1,p,n}$ is calculated assuming a circular panel clamped along its edges [41].

It can be seen from Figure 4b,e,h and Figure 5a–c that the limp membranes dampen the dissipation peaks that contribute to the pass-band of the micro-perforated acoustic fishnet, as long as $a/\Lambda < 0.065$ ($a < 0.5$ mm) and $\Lambda/\lambda < 0.05$. It can be shown from Equations (15) and (16) that the inertial membrane impedance $Z_{m,n}$ contributes to increasing the overall specific resistance of $Z'_{\mathrm{MPP},n}$, thereby broadening and merging the first pass-band resonances, as seen in the blue curves of Figure 5. Meanwhile, the dissipation, absorption and transmission peaks are also dampened. However, for perforated acoustic fishnets ($a/\Lambda > 0.065$), Figure 4b,e,h and Figure 5d–f show that the limp membranes have minute effects whatever $\Lambda/\lambda$, due to the already low resistances of the rigid perforates with a large perforation ratio.

As for the resonating MPPs, Figure 4c,f,i and Figure 5 show that the panels first mode produces a narrowband dissipation peak around 800 Hz whatever $a/\Lambda$, associated with a sharp absorption peak and transmission dip. However, this volumetric mode has different effects over the pass-band whether the fishnet is made up of microperforates or perforates. As for microperforates ($a/\Lambda < 0.065$, $a < 0.5$ mm), the resonance frequency at 800 Hz falls within the half-bandwidths of the HC acoustical resonances gathered in the pass-band. It results in a strong coupling with these resonances. It upshifts (resp. downshifts) the HC resonances above (resp. below) 800 Hz as it decreases (resp. increases) their acoustic reactance. As a result, accounting for the panels' first volumetric mode tends to enlarge the width of the pass-band. It also lowers (resp. increase) the acoustic resistance of the HC resonances located above (resp. below) $f_{1,p,n}$. This is accompanied by decreases (resp. increases) in the dissipation and absorption peaks, as seen from Figure 5a,b. Figure 5c shows that it worsens the partition's insulating performance above $f_{1,p,n}$. As for perforates ($a/\Lambda > 0.065$, $a > 0.5$ mm), the panels' volumetric mode weakly cross-couples with the neighboring HC resonances, very widely spaced in terms of frequency. Figure 5d–f shows that it still contributes to the acoustical performance, as seen through a sharp peak at $f_{1,p,n}$ but with a dampening and cross-coupling effect on the HC resonances the lower that $a/\Lambda$ increases. It can be seen from Figure 5 (red curve) that the above effects at work for micro-perforates are negligible for macro-perforates.

## 4. Acoustic Performance of Multi-Layered Functionally Graded (M)PPs

It was shown in Section 3 that imposing a constant radius of the holes and perforation ratio for all the panels through the partition ensures the occurrence of pass-bands and stop-bands. However, the stop-band occurs at quite high frequencies (above 3 kHz, as seen in Section 3 for $a = 0.3$ mm). The pass-band is made up of absorption peaks induced by the HC resonances, which may be merged or shifted due to vibration effects. As such, the acoustical performance of acoustic fishnets appears to be moderate at low–mid frequencies below 1 kHz. A better performance could be obtained over this frequency range if a proper selection for the radius of the holes was made for each panel constituting the partition. A key idea would be to achieve both impedance matching and a high visco-thermal dissipation as the incident wave enters the partition through the holes. This could be obtained by gradually decreasing the radius of the holes across the partition, following a chirped distribution that could be optimized for any of the constitutive MPPs. However, a constraint would be to maintain a subwavelength overall length for the partition, thus leading to a compact multi-layered FGP. This objective will be carried out through an

optimization study that takes into consideration both the transmitted and reflected powers to ensure the maximum dissipation of the incident power inside the partition.

### 4.1. Broadband Optimization of FGPs

An optimal selection for the radius of the holes across the FGP is a combinatorial optimization problem where all the physical parameters are cross-related, in the sense that a variation in any of the parameters may significantly affect the others. The selected cost function, the frequency-averaged dissipation in the low-frequency range, can also present many sub-optimal maxima, and classical gradient-based optimization algorithms take the risk of being trapped in these non-optimal solutions. It is then necessary to use other optimization techniques, such as natural algorithms, a special class of global metaheuristic optimization processes, that reproduce the physical selection mechanisms present in real life. In this work, we use the simulating annealing (SA) optimization method [42] to find the optimal distribution of the holes' radius that globally maximizes the total sound power dissipated by the FGP between 10 Hz and 1 kHz.

The fixed parameters for the FGP are the panel thickness, set to $d_{p,n} = 0.5$ mm, the pitch of the holes, set to $\Lambda_n = 20$ mm, and the overall thickness of the system, set to $L = 300$ mm, which stays at a subwavelength up to 1150 Hz. This leads to an air gap thickness $d_{g,n} = 15.3$ mm. Twenty perforated panels were used, assuming a monotonically decreasing chirped distribution for the radius of the holes across each layer, such as:

$$a_{n+1} = a_n - \alpha_n \frac{\Lambda_n}{2}, \tag{19}$$

with $a_1 = \Lambda/2.5$ and $\alpha_n > 0$ the chirp parameter being optimized.

Figure 6a presents the dissipation of the rigid partition over the optimization bandwidth for several distributions selected for the radius of the holes while the SA algorithm was running (shown as grey curves). It can be seen that wide differences can be found in the performance depending on the decay for the radius of the holes. The thick black curve corresponds to the optimal dissipation results, which maximize the average dissipation of the partition over the targeted frequency range. It leads to a broadband dissipation coefficient that stays above 0.7 between 88 Hz and 990 Hz, with perfect dissipation at 820 Hz ($L/\Lambda = 0.72$). On the other hand, the thin black curve corresponds to a distribution for the radius of the holes that provides the worst performance.

The distributions associated with both extreme performances are presented in Figure 6b along the axial length of the partition. The best and worst solutions start with the same radius for the holes over the front perforated panel ($a_1 = 8$ mm), but the differences between both distributions of radius become all the more important as one progresses towards the partition's transmitting side. The worst solution is conducive to a fully perforated FGP ($a_{20} = 0.9$ mm), whereas the best solution exhibits an almost linear decay for the radius of the holes across the partition, with only the four last panels being micro-perforated (down to $a_{20} = 0.2$ mm). Thus, a proper selection of the decay rate for the radius of the holes in FGPs plays a crucial role that makes the optimization procedure unavoidable when tailoring the FGP noise control performance towards specific frequency bands.

Figures 6a and 7a–c show that the worst solution leads to a low dissipation performance induced by very few isolated HC resonances activated up to 1 kHz, whereas the best solution enables the grouping and merging of these resonances over the targeted frequency range (except the first HC resonance at 110 Hz), thereby broadening the acoustical performance of the FGP in the low–mid frequency range. Figure 7b,c show that the absorption coefficient does not fall below 0.8 over this bandwidth, while the TL stays between 8 dB and 35 dB. This is reminiscent of the performance of Acoustic Black Holes [43], another category of FGP with graded cavity depths, that trap and dissipate incident waves over a large bandwidth. As expected, the dissipation shown in Figure 7 decays above 1 kHz. This is accompanied by a decrease in the absorption, but the TL still increases with a frequency above 1 kHz.

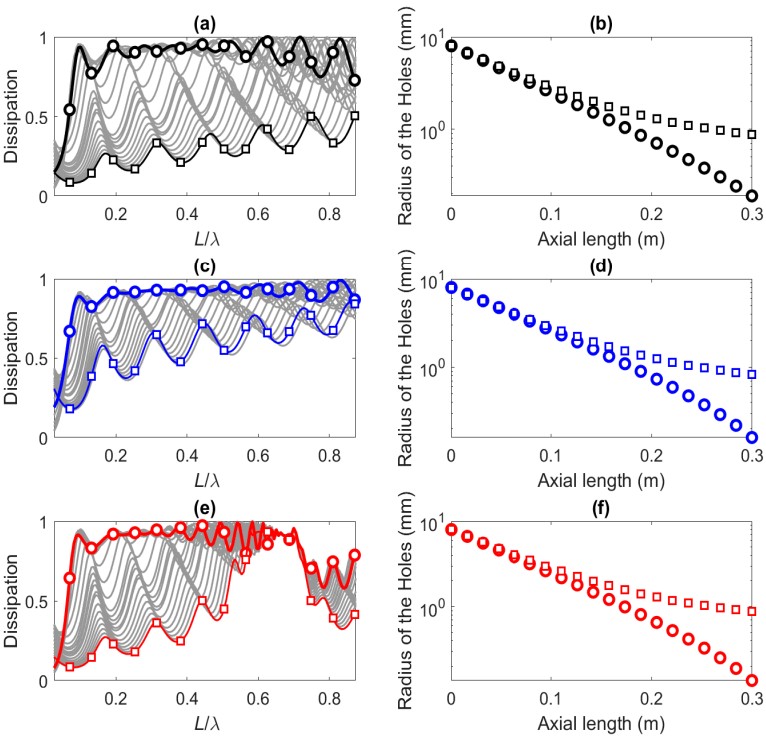

**Figure 6.** Dissipation spectra (**left**) and variation of radius of the holes (**right**) associated with maximization (circles) and minimization (squares) of the total power dissipated up to 1 kHz by a 19-layer micro-perforated FGP using a simulated annealing algorithm and assuming rigid panels [(**a**,**b**), black]; elastic limp membranes [(**c**,**d**), blue]; and resonant vibrating panels [(**e**,**f**), red]. Grey curves in (**a**,**c**,**e**) show the intermediate dissipation spectra calculated during the optimization process.

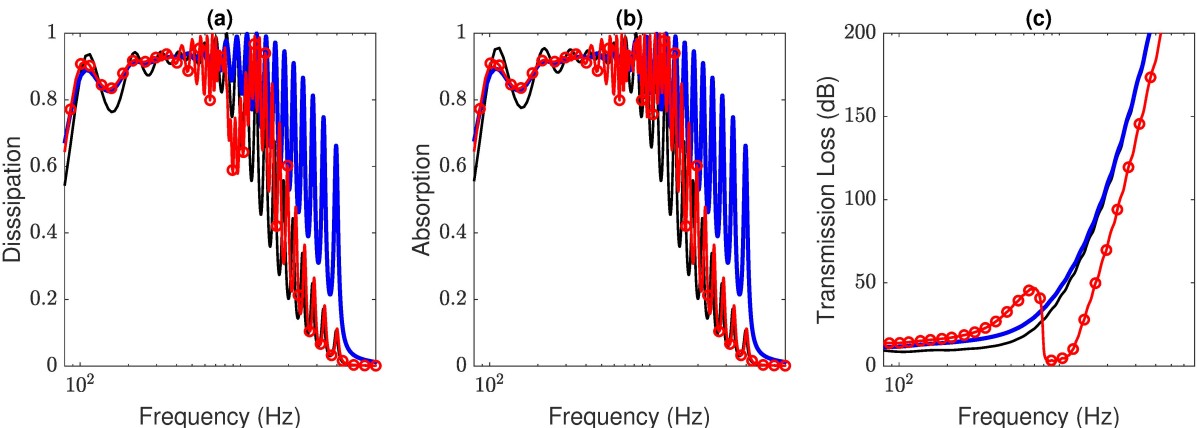

**Figure 7.** Dissipation (**a**); absorption (**b**); and transmission loss (**c**) spectra of an optimized 19-layer micro-perforated FGP assuming rigid panels (black), elastic limp membranes (blue), and resonant vibrating panels (red with circles).

### 4.2. Vibrational Effects on the Broadband Optimization of FGPs

SA optimization is performed cost-efficiently by accounting for the limp (micro-)perforated membranes and resonant vibrating panels in the ITM. As already observed in Section 3, the membrane inertial impedance, $Z_{m,n}$, contributes to increasing the overall specific resistance, $Z'_{\text{MPP},n}$, of the (micro-)perforated membrane. As shown in Figure 6c, this effect broadens the individual half-bandwidth resonances and favors the merging of the first eight HC resonances up to 1 kHz ($L/\lambda = 0.88$). It also smooths out the dissipation and absorption ripples over the efficiency bandwidth up to 1 kHz. It provides at least an

equal or better performance in both dissipation, absorption, and transmission with respect to the rigid optimal FGP, as seen when comparing the blue and black curves in Figure 7.

The optimization of FGPs with all the panels vibrating on their first volumetric mode (resonant at 800 Hz), produces in Figures 6e and 7 (red curve) a high plateau of dissipation (greater than 0.9), with ripples occurring towards 800 Hz followed by a dissipation peak and then a drop in dissipation above 800 Hz. This is accompanied by a drop in TL (or a large transmission) that can be seen in Figure 7c between 800 Hz and 1200 Hz, but the absorption is comparatively not so impeded by the structural resonance. This is expected since the absorption is essentially dependent on the front panel resonance, whereas the transmission accounts for all the resonances of the panels through the partition. Therefore, the effect of the resonant panels in FGPs is to reduce the efficiency ranges of the dissipation and the transmission up to $f_{1,p,n}$, the panels first resonance frequency induced by its volumetric mode. Assuming lower values of $f_{1,p,n}$ further reduces the FGP efficiency range and leads to a similar scenario: a high-valued dissipation plateau, as well as a high TL, with an accumulation of merged resonances up to $f_{1,p,n}$, followed by a drop in dissipation and TL above $f_{1,p,n}$. Therefore, $f_{1,p,n}$ is an upper bound for optimizing FGPs whose MPPs resonate at the same frequency, $f_{1,p,n}$. Alternatively, $f_{1,p,n}$ could be considered as a design variable, up to which the optimization of FGPs provides a high performance.

It is of interest to note that the optimal FGP performance is achieved for very similar decay rates for the radius of the holes, as observed when comparing the distributions of the circles in Figure 6b,d,f. The differences are lower than 0.1 mm (resp. 0.05 mm) between the optimal decay rates for FGPs made up of elastic membranes (resp. resonant MPPs) with respect to rigid FGPs. This shows how the optimal decay rate for the radius of the holes is robust against vibrational effects.

### *4.3. Critical Coupling Analysis*

Further insights can be gained on the relationship between the losses and the acoustical performance of optimized FGPs if one examines the zero-pole placement of the eigenvalues and eigenvectors of the scattering matrix in the complex frequency plane. Each pair of zero-poles is associated with a resonance of the system. In the lossless case, assuming $\text{Real}\left(Z_{\text{MPP},n}\right) = 0$ in Equations (2) and (3) together with $\xi_{1,p,n} = 0$ in Equation (18), the system is purely reactive and the zeros and poles are located symmetrically with respect to the real frequency axes, as observed in [9] for lossless Helmholtz resonators. When losses are introduced, the zeros and poles are vertically downshifted in the complex frequency plane. Figures 8b, 9b and 10b show that some zeros of the eigenvalues (that appear as circled black points) cross the real axis. As explained in Section 2.2, because the eigenvalues $\lambda_{1,2}$ of the **S** matrix are zero-valued at these points, a critical coupling condition (CCC) is achieved for these associated resonances given the amount of visco-thermal losses in the FGP.

### 4.3.1. Optimization of Rigid FGPs

In the case of FGPs with rigid panels, the optimization process led to the broadening and merging of the first eight HC resonances. The eighth HC resonance (located by circles in Figure 8) reaches the CCC with a unit dissipation peak and $|\lambda_{1,2}| = 0$ at 800 Hz. Moreover, Figure 8c shows that both components, $R^-$ and $\sqrt{R^- R^+}$, of the first eigenvector associated with $\lambda_1$ become zero-valued at the same frequency, as they involve $R^- = 0$ at this frequency, indicating that all the incident waves coming from the left enter the optimized FGP without reflection ($R^- = 0$) nor transmission ($T = 0$, see Equation (10)) at this frequency. Moreover, Figure 8d shows that only the second component, $\sqrt{R^- R^+}$, of the second eigenvector associated with $\lambda_2$ becomes zero-valued at the same frequency (because $R^- = 0$), while the other component $R^+$ is not zero-valued. This means that the CCC is not satisfied for an incident wave entering the optimized FGP from the right, as should be expected, since the partition is not mirror symmetric and has only been optimized for an incident wave impinging from the left side.

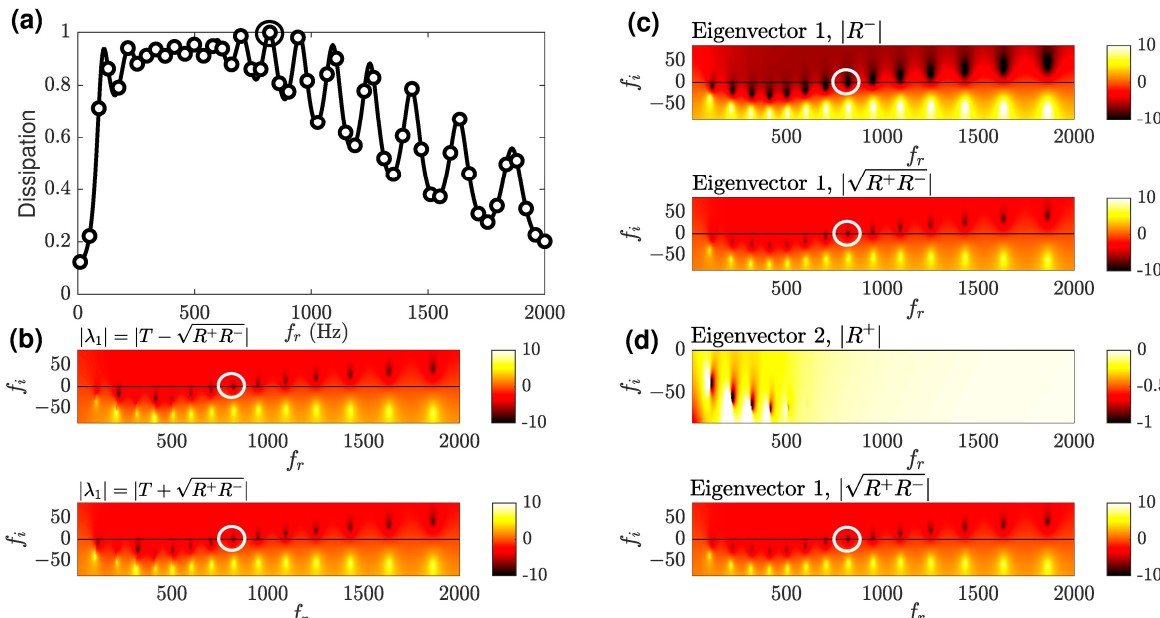

**Figure 8.** (**a**) Dissipation spectrum of an optimized 19-layer micro-perforated FGP assuming rigid panels; (**b**) modulus (log10 values) of the S-matrix eigenvalues in the complex frequency plane; (**c**) modulus (log10 values) of the eigenvector components associated with $\lambda_1$ in the complex frequency plane; and (**d**) modulus (log10 values) of the eigenvector components associated with $\lambda_2$ in the complex frequency plane. Circles spot the frequency location at which the critical coupling condition occurs.

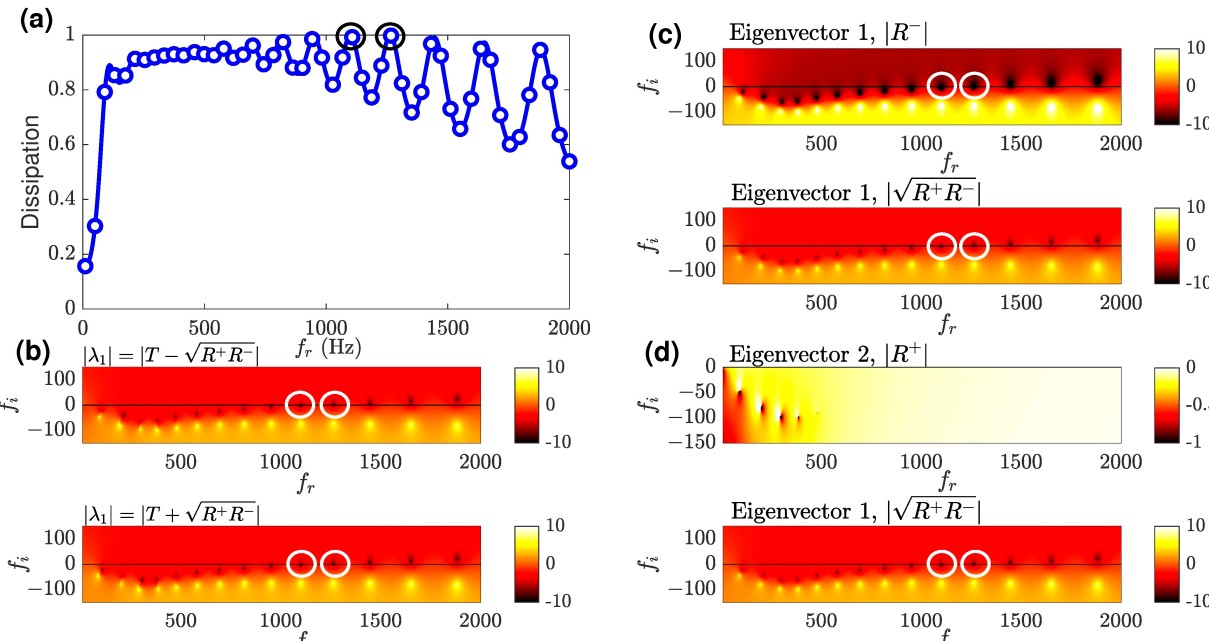

**Figure 9.** (**a**) Dissipation spectrum of an optimized 19-layer micro-perforated FGP assuming elastic limp membranes; (**b**) modulus (log10 values) of the S-matrix eigenvalues in the complex frequency plane; (**c**) modulus (log10 values) of the eigenvector components associated with $\lambda_1$ in the complex frequency plane; and (**d**) modulus (log10 values) of the eigenvector components associated with $\lambda_2$ in the complex frequency plane. Circles spot the frequency locations at which critical coupling conditions occur.

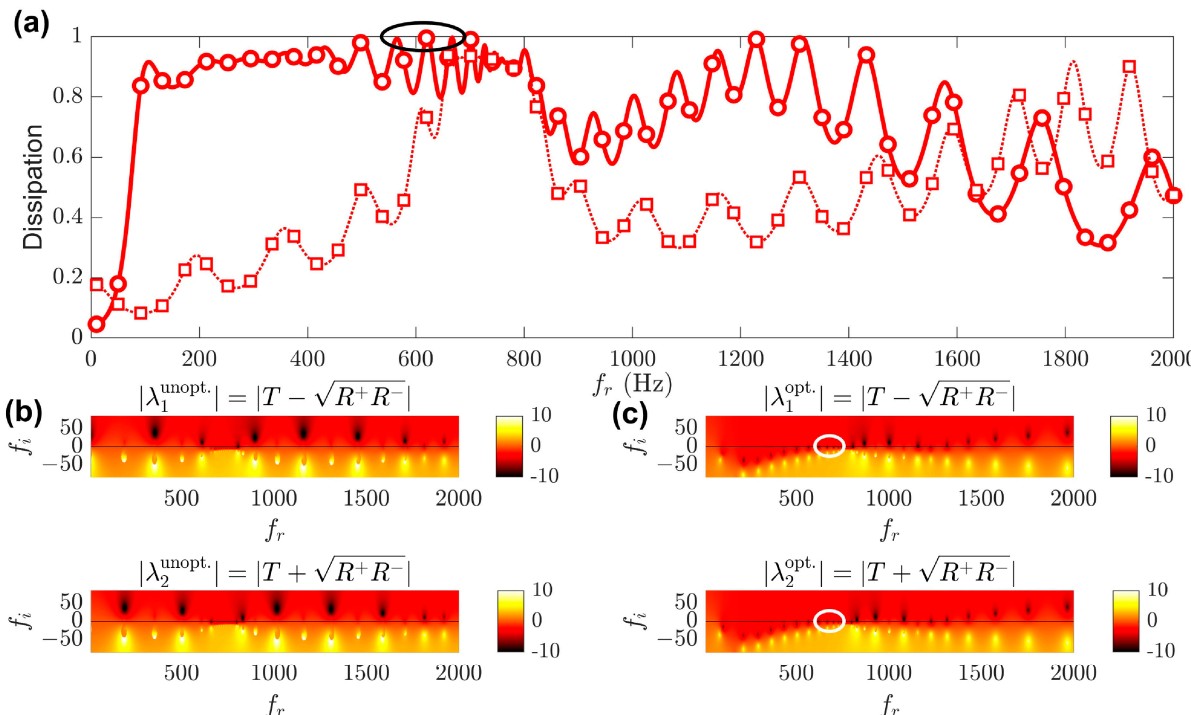

**Figure 10.** (**a**) Dissipation spectra of optimized (plain red with circles) and not-optimized (dottted red with squares) micro-perforated FGPs made up of 19 layers and assuming resonant vibrating panels; (**b**) modulus (log10 values) of the S-matrix eigenvalues of the standard FGP in the complex frequency plane; and (**c**) modulus (log10 values) of the S-matrix eigenvalues of the optimized FGP in the complex frequency plane; Ellipses spot the frequency locations at which critical coupling conditions occur.

It can be seen from Figure 8b that the optimization of the rigid FGPs led to a redistribution of the HC resonances in two classes: those in an over-resistive regime (the first seven HC resonances) located in the optimization bandwidth below 800 Hz, such that $f_i < 0$ when $|\lambda_{1,2}| = 0$, and those in an under-resistive regime, such that $f_i > 0$ when $|\lambda_{1,2}| = 0$. The first set of resonances is too lossy, whereas the second set is not lossy enough. Adjusting a proper amount of losses for each resonance would be a strategy that would enable setting all the zeros over the real frequency axis and achieving perfect dissipation over the whole bandwidth.

### 4.3.2. Optimization of FGPs with Elastic Limp Membranes

Figure 9 shows that the optimization of the FGP with elastic limp membranes brings added resistances to the MPPs and leads to further broadening and merging of the first 11 HC resonances over a larger bandwidth up to 1250 Hz. It extends beyond 1 kHz, the upper frequency bound of the optimization range, with the tenth and eleventh HC resonances satisfying the CCC and those above 1250 Hz being in an under-resistive regime. Note that the y-scales are different between Figures 8b and 9b, so that the zero-pole locations are further downshifted in Figure 9b with respect to Figure 8b, due to the added resistance by the limp micro-perforated membranes. As in the rigid case, only the first eigenvector components vanish when the CCC is satisfied, as observed in Figure 9c,d.

### 4.3.3. Optimization of FGPs with Thin Resonant Vibrating Panels

Accounting for the panels' volumetric resonant mode results in a significant redistribution of the FGP resonances in the complex frequency plane, both before and after optimization. Before optimization, one observes from Figure 10b a regular set of under-resistive HC resonances associated with moderate dissipation peak values below 0.5, except

between 660 Hz and 800 Hz, where a plateau of high dissipation values (0.93) occurs. Note that, unlike in the previous cases, one does not spot a one-to-one relation between zeros and poles, since extra-poles are interlaced between consecutive zero-pole pairs. In the high dissipation zone, strong coupling between the panels' volumetric mode and the nearby HC resonances led to multi-resonance splitting. One observes in Figure 10b an accumulation of over-resistive elasto-acoustic panel-HC (P-HC) resonances from 660 Hz up to the panels resonance frequency, $f_{1,p,n} = 800$ Hz, with pairs of zero-poles getting closer to the real axis as the frequency increases towards $f_{1,p,n}$. Out of this frequency zone, since the holes are super-millimetric for all the panels (see square markers in Figure 6f), weak cross-coupling occurs between the panels' (P) structural mode and the HC resonances, as seen in Section 3.2.

After optimization, Figure 10c shows that this narrowband accumulation of P-HC resonances spreads out towards the low frequencies, due to the enhanced cross-coupling over a larger frequency band between the P and HC resonances enabled by the micro-perforates (see Section 3.2). Moreover, the obtained high dissipation values are favored by the stiffness-like behavior of the vibrating panels impedance when $f < f_{1,p,n}$. The optimization process thus redistributes the P-HC resonances below $f_{1,p,n}$ and modifies the distribution for the radius of the holes to favor the CCC at these resonances, as if the FGP was made of rigid panels. Above 800 Hz, a regular distribution of zero-pole pairs associated with under-resistive HC resonances re-appears in Figure 10c, with a couple of peaks near 1250 Hz close to near-unit dissipation. Although not shown, once again, only the first eigenvector components reflect the CCC that only occurs under left-side excitation before and after optimization.

## 5. Discussion

In this work, the vibrational effects of two strategies were assessed on the acoustical performance of either periodic (acoustic fishnet) or functionally graded (micro-) perforated partitions, whether the (micro-) perforated solid parts were made of elastic limp membranes or thin panels resonant within the partition efficiency range.

The use of (micro-) perforated elastic membranes added a sufficient amount of resistance that broadened and merged the dissipation, absorption, and transmission peaks belonging to the pass-band of micro-perforated acoustic fishnets. It also smoothed out the ripples due to HC resonances over the optimization bandwidth of the FGPs, thereby achieving almost constant high dissipation values, as well as low transmission and high absorption over a broad bandwidth from $L/\Lambda = 0.2$ up to $L/\Lambda = 0.6$. Compared to the optimized rigid FGP that led to a total frequency-averaged dissipation of 0.9 over the optimization bandwidth (10 Hz–1 kHz), a similar total dissipation of 0.92 was achieved when using limp membranes while avoiding dissipation dips. They can be made of ultrathin polymer (micro-)perforated membranes. They comply very well with the lightweight and cost-effective production of acoustic metamaterials, designed to be efficient over a broad low–mid frequency range. However, their robustness towards severe environments (high temperatures, high-speed flow, and oxidant gases, etc.) still requires further research in terms of the material resilience properties.

Conversely, using thin (micro-perforated) panels is a more robust option that can withstand harsh conditions, albeit less lightweight than limp membranes. Because of their sub-millimetric thickness, their first resonant volumetric mode likely falls within the low-frequency range of the efficiency of acoustic fishnets or FGPs. On top of generating a narrowband dissipation peak, the first volumetric mode of MPPs strongly coupled with the neighboring HC resonances, leading to a new set of elasto-acoustic (P-HC) resonances. Such vibrational effect extended the efficiency bandwidth of the micro-perforated acoustic fishnets as long as $a/\Lambda < 0.065$. The resonance frequency of the first mode, $f_{1,p,n}$, also set an upper limit to the broadband optimization of FGPs, up to which a high dissipation, high absorption, and low transmission could be achieved, but beyond which, these acoustic indicators exhibited a moderate performance. A total frequency-averaged dissipation

of 0.86 was achieved up to 1 kHz from an optimized FGP with thin MPPs resonant at $f_{1,p,n} = 800$ Hz. It therefore provides a robust subwavelength dissipative solution, whose bandwidth can be tuned by a proper choice of $f_{1,p,n}$.

A critical coupling analysis revealed how the membranes or panel vibrations redistributed the locations of the HC resonances, as well as their cross-coupling with the panels' first volumetric mode. It also provided insights on the amount of losses required to make each resonance critically coupled over the optimization bandwidth. In particular, such losses could be tailored by specifying the amount of damping brought about by visco-elastic micro-perforated panels [14] constitutive of the partition.

The thin micro-perforated panels involved in the partitions can be produced from additive manufacturing, such as stereolithography or laser sintering processes, suitable for producing the smallest diameter for the holes required by the optimized FGPs, e.g., down to 0.26 mm. Note that the optimized FGP partitions appeared to be robust to small uncertainties induced by the manufacturing process (typically 0.05 mm for stereolithography) onto the required values for the diameter of the holes across the partition.

Further works should focus on experimentally testing the vibrational effects of limp membranes or thin resonating panels on optimized acoustic fishnets or FGPs under normal, but also under grazing sound incidences, to show the practical efficiency of these subwavelength absorbers, not only in building acoustics, but also as lightweight and compact silencing wall treatments in duct acoustics.

## 6. Conclusions

This study analyzed how vibrational effects, often neglected, may hinder the wideband acoustical performance of sound-absorbing partitions, made up of a distribution of either periodic or graded (micro-)perforated membranes or panels. It was found that the elasticity behavior of the thin membranes had a beneficial effect on the broadband acoustical efficiency of the periodic or graded partitions. It merged and smoothed out the contribution of the individual acoustical resonances, thereby producing near-constant high dissipation and absorption values, as well as low transmission, over a wide frequency range.

Partitions made up of micro-perforated panels are a more robust design, but the panels' volumetric modes may impede their absorption properties. It was observed that the first volumetric mode of the panels cross-coupled with the acoustical resonances of the partition and redistributed their spectral locations. Hence, this structural resonance frequency set an upper limit, below which the partition could achieve a broadband performance, e.g., a high dissipation, high absorption, and low transmission. This upper frequency bound should be accounted for when setting the frequency limits of the total dissipation to be optimized.

**Author Contributions:** Conceptualization, T.B. and C.M.; methodology, C.M. and T.B.; software, C.M., validation, T.B.; formal analysis, C.M. and T.B.; investigation, T.B. and C.M.; resources, T.B. and C.M.; data curation, T.B.; writing—original draft preparation, T.B. and C.M.; writing—review and editing, C.M.; visualization, C.M.; supervision, T.B.; project administration, T.B. and C.M.; funding acquisition, T.B. and C.M. All authors have read and agreed to the published version of the manuscript.

**Funding:** This work is part of the project TED2021-130103B-I00, funded by MCIN/AEI/10.13039/501100011033 and the European Union "NextGenerationEU"/PRTR. It has also received support from the French government under the France 2030 investment plan, as part of the Initiative d'Excellence d'Aix-Marseille Université—A*MIDEX (AMX-19-IET-010).

**Data Availability Statement:** Data supporting the reported results are available on request.

**Conflicts of Interest:** The authors declare no conflict of interest.

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
