# Peer review of "Vibrational Effects on the Acoustic Performance of Multi-Layered Micro-Perforated Metamaterials"

_vibration, doi:10.3390/vibration6030043_

Round 1

Reviewer 1 Report

This is an interesting paper and well prepared. Some minor remarks on this:

Line 65: It is written: ... typically 17 kHz in [2].
Delete "in". It should read: ... 17 kHz [2].

Line 155: It is written: Maa’s formulation [15] for the effective transfer impedance of the nth (M)PP takes the form ...Equation (1)

Equation (1) is not understood. It's hard to follow the reasoning. There is no effective transmission impedance in Maas's work refered as [15]. Could you be more detailed or provide a supplementary explanation or extension of the derivation of the equation? This would help to understand the results in more detail.

Author Response

Please, see document attached

Reviewer 2 Report

This paper conducted a theoretical analysis of the vibrational and acoustical characteristics of multi-layered micro-perforated metamaterials. This paper is novel and well-written. The analysis in the paper is very detailed, and the quality of the research results presented is also very high. I recommend this paper be published after a minor revision.

1. The vibrational and acoustical characteristics of multi-layered micro-perforated metamaterials are a hot topic in the research field of sound and vibration. Therefore, there must be a lot of literature on this aspect recently. Can the authors add several new literature from recent years to the Reference?

Author Response

Please, see document attached

Reviewer 3 Report

The paper discusses the simulations' results for different types of multi-layer metamaterials in two options – MPP and FGP. The authors compared the modeling results with and without the limp membrane behavior and discussed the influence of this effect on final metamaterial performance. I believe the research quality is proper, and the topic is novel enough to qualify for publication. I overlooked the scientific issues and mistakes across the paper, but ultimately, my decision qualifies it for major revision as the paper has many technical issues and is very difficult to read. The range of the required revision exceeds the minor revision scope, so my decision is significant, but if all issues are solved, I expect the switch to acceptance as the scientific quality is high. Please find the major issues I have detected in the review process:

1. The big problem is the English language. I will point out some direct cases in the next section, but I wanted to emphasize that the whole paper should be reviewed for better English, as I will not do this in reviewing process. I do not insist that a native speaker or technical editor is required, but the paper has so many obvious issues that even basic automatic editor such as Grammarly will significantly improve here.

2. The introduction (pages 2 and 3) is quite long, while I find most of those papers cited there and described as non-directly connected with the paper's topic. Only lines 115-120, with references 12-13, are direct references to the essential paper. I would suggest shortening the introduction and focusing on the essential references to make the paper better organized and easier to read.

3. In Chapter 2, I found a big lack of references and precision regarding the essential input from authors and what is used from the previous work.

4. The equations formulation – this requires grand rework. I understand that the authors did not want to use every equation as a separate line, but this form requires rework as now it is challenging to follow the pipeline and reefer to given equations, see example page 4 and later. The authors should work on this:

a. Use only essential equations

b. Use the numbering and separate lines for single equations with fractions and roots.

c. Take care of proper equation description with technical English.

5. Chapters 3 and 4 – I see a lack of consistency in the presentation of the results, which makes the paper very difficult to read and understand. If the goal is to compare several options for metamaterials modeling, the authors should stick with some form of presentation and the same type of plots for a different design to allow their direct comparison, while in all chapters for both MPP and FGP, they are mixed. The most accessible and user-friendly would be the presentation of absorption for different options; while I do not know why there are eigenvalues in Figures 8,9, and 10, this is very difficult to understand. I suggest a grand rework here.

Minor direct-to-text remarks:

1. Lines 39-40 and some later – I am afraid I have to disagree that we must also work with sound absorption, not insulation, in designing the insulation wall. Those are different materials with different functions. Also, the authors should work with the text to avoid further mixing of sound insulation and absorption

2. Lines 66-67 – this sentence needs to be rewritten

3. Lines 134-135 – I do not understand this sentence

4. The first paragraph of chapter 3.1 needs to be rewritten. Totally not clear

5. Chapter 4.3 – I do not understand this part and why It is included in the text

6. I think the shortcut MPP is recognized and viable in the state of the art. There is no need for (micro) or (M)PP – just MPP.

7. When the transmission and reflection coefficients are mentioned, I suggest using absorption to this set instead of dissipation, as this may be easier to follow and more recognized across the references.

I see a lot of English errors across the text. The paper should be rewritten using automatic text correction software, as the errors are obvious and easy to track. See essential language errors I have wanted to point out, but keep in mind that this is not the English correction review, so the authors should work with the whole text:

Lines 32-35 – long sentence with mixed sense

Line 45 – wrong sentence start

Lines 55-56 "here included"

Lines 109-11 "build upon the idea" "whose filling fraction"

Lines 159-166 – this is one sentence for 8 lines

Linee 172 – jump of pressure?

Line 193 and 195 – "no wave going out of the system"

Multiple typo – "hole radii"

Those errors are continued in the later part of the paper, but this reviewer decided not to continue this language revision.

Author Response

Please, see document attached

Round 2

Reviewer 3 Report

The Authors reviewed the text according to my comment from the previous review round. In the current state, I am satisfied with the changes applied, and I believe the paper should be accepted for publication. The current editing style has multiple formatting errors – different fonts and spacing between rows across the paper – but they can be improved in the final print-ready version preparation stage.

Author Response

Please, see document attached
